# LTGS: LONG-TERM GAUSSIAN SCENE CHRONOLOGY FROM SPARSE VIEW UPDATES

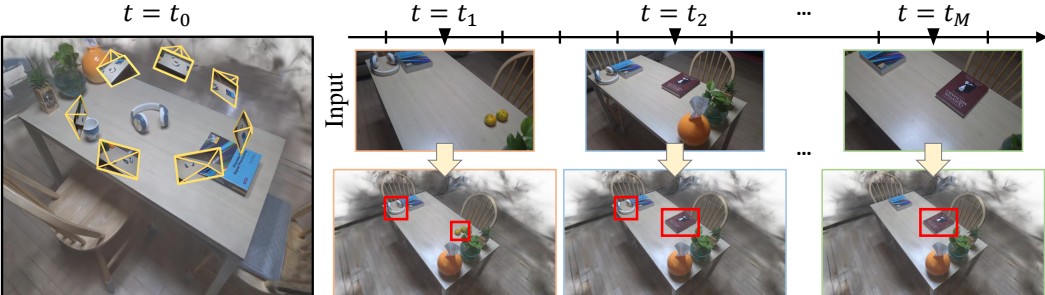

Figure 1: **We introduce LTGS to efficiently update the Gaussian reconstruction of the initial environments.** Given the spatio-temporally sparse post-change images, our framework tracks object-level changes in 3D and enables modeling scenes with long-term changes.

## ABSTRACT

Recent advances in novel-view synthesis can create the photo-realistic visualization of real-world environments from conventional camera captures. However, acquiring everyday environments from casual captures faces challenges due to frequent scene changes, which require dense observations both spatially and temporally. We propose long-term Gaussian scene chronology from sparse-view updates, coined **LTGS**, an efficient scene representation that can embrace everyday changes from highly under-constrained casual captures. Given an incomplete and unstructured Gaussian splatting representation obtained from an initial set of input images, we robustly model the long-term chronology of the scene despite abrupt movements and subtle environmental variations. We construct objects as template Gaussians, which serve as structural, reusable priors for shared object tracks. Then, the object templates undergo a further refinement pipeline that modulates the priors to adapt to temporally varying environments based on few-shot observations. Once trained, our framework is generalizable across multiple time steps through simple transformations, significantly enhancing the scalability for a temporal evolution of 3D environments. As existing datasets do not explicitly represent the long-term real-world changes with a sparse capture setup, we collect real-world datasets to evaluate the practicality of our pipeline. Experiments demonstrate that our framework achieves superior reconstruction quality compared to other baselines while enabling fast and light-weight updates.

## 1 INTRODUCTION

With recent advances in novel-view synthesis, such as Neural Radiance Fields (NeRFs) (Mildenhall et al., 2020) or 3D Gaussians Splatting (3DGS) (Kerbl et al., 2023), a casual user can reconstruct a 3D environment using a conventional camera input, and enjoy a photo-realistic experience of exploring the environment. The representation stores the complex distribution of light and geometry of the static scene in an unstructured format. If we model the everyday environments where people live, daily activities often induce changes within the scene, quickly making the reconstruction obsolete. One may need to rerun these algorithms from scratch, or incorporate 4D representations that encapsulate the dynamic movements of the 3D representation (Pumarola et al., 2020; Wu et al., 2024;

Gao et al., 2022). The former discards the previously acquired information, while the latter requires smooth motions to be captured and processed from continuous observation. Both approaches suffer from significant redundancy and are not desirable for modeling everyday environments in practical applications, such as location-based services, digital twins, or robotic setups.

We argue that a practical strategy for modeling evolving real-world environments is to efficiently detect and update changes. Instead of requiring continuous captures of the entire scene, we suggest a light-weight update of the changed region from a sparsely captured image, as demonstrated in Figure 1. While the setup is designed to be a plausible approach to model everyday environments, it is highly under-constrained and therefore requires a strong scene prior. The challenge is to efficiently update the scene without many artifacts while maintaining the necessary information for photorealistic rendering. Previous works introduce approaches that use only a sparse set of input images for novel-view synthesis (Niemeyer et al., 2022; Guangcong et al., 2023; Fan et al., 2024), which still suffer from severe artifacts when viewpoint changes significantly. The continual learning method can sustain the pre-captured information (Wu et al., 2023; Ackermann et al., 2025; Zeng et al., 2025) only with multiple captures for updates, as it lacks structural priors.

We propose an integrated pipeline that detects and updates Gaussian splatting scene representations in environments with diverse object states from sparse observations. We refer to the framework as long-term Gaussian scene chronology from sparse view updates, or LTGS. Real-world scenes can undergo multiple types of scene changes, including variations in geometry, appearance, or lighting. Recent works demonstrate that the subtle appearance and lighting changes can be partially addressed by adding learnable embeddings or auxiliary neural networks (Martin-Brualla et al., 2021; Lin et al., 2023; Kulhanek et al., 2024; Lin et al., 2024). In this work, we focus on **object-level changes**, which involve abrupt geometric alterations such as insertions, removals, replacements, or relocations, providing a structural mechanism to efficiently account for the consequences of everyday interactions.

Given an initially reconstructed 3DGS without any segmentation, we need to robustly extract the object-level structure, which confines the granularity of change estimation under ambiguous observations. Our scene update involves object tracking, relocalization, and reconstruction for individual objects. Combining multiple image-space evidences of segmentation and feature extraction, we detect and distill the change to a 3D representation. We then aggregate the observations to build an object-level Gaussian template that models an object shared across time. The template serves as a **reusable 3D prior** to relocalizing objects at different times, overcoming the ambiguities in the sparse views. Then we can reiterate the aggregation step such that the images of multiple time spans refine the shared template to best explain the overall observations via simple transformations. To evaluate our framework in practical scenarios, we captured real-world datasets containing multiple shared objects in various layouts, with few-shot observations spanning multiple time steps. Our pipeline demonstrates robust performance in the challenging setup where previous approaches struggle.

Our key contributions are summarized as follows:

- We address the problem of updating an initial 3DGS reconstruction in a highly efficient manner by using a set of spatio-temporally sparse images capturing long-term changes.
- We present LTGS, an integrated strategy to track, associate, and relocalize the objects, and reconstruct the evolving scenes.
- We propose a new real-world dataset, casually capturing environments with dynamic object-level changes across multiple timesteps to evaluate our framework.

## 2 RELATED WORKS

We build on Gaussian splatting representation and propose a novel yet challenging practical setup that allows lightweight updates of temporal changes from sparse view inputs. The setting partially shares important properties with recent variations of novel-view synthesis that enable temporal extensions or few-shot reconstruction, as summarized in Table 1.

### 2.1 PHOTOREALISTIC RECONSTRUCTIONS IN NON-STATIC SCENES

Several works successfully extend NeRFs (Pumarola et al., 2020; Li et al., 2022) or 3DGS (Luiten et al., 2024; Wu et al., 2024; Wang et al., 2025) to model dynamic scenes. These works require dense

Table 1: **Related methods comparison.** Our method captures abrupt geometric changes without requiring continuous motion and maintains reconstructions of multiple timesteps using a decomposable geometric scene prior that is reusable, thus allowing fast updates from a sparse set of images.

| Method | Discont. motion | Temporal recon. | Few-shot | Reusable prior | Training speed |
|---|---|---|---|---|---|
| 3DGS (Kerbl et al., 2023) | ✗ | ✗ | ✗ | ✗ | Fast |
| InstantSplat (Fan et al., 2024) | ✓ | ✓ | ✓ | ✗ | Fast |
| 4DGS (Wu et al., 2024) | ✗ | ✓ | ✗ | ✗ | Moderate |
| NSC (Lin et al., 2023) | ✗ | ✓ | ✗ | ✗ | Slow |
| 3DGS-CD (Lu et al., 2025) | ✓ | ✗ | ✓ | ✓ | Fast |
| CL-NeRF (Wu et al., 2023) | ✓ | ✗ | ✗ | ✗ | Slow |
| CL-Splats (Ackermann et al., 2025) | ✓ | ✗ | ✗ | ✗ | Fast |
| LTGS (Ours) | ✓ | ✓ | ✓ | ✓ | Fast |

input observations both temporally and spatially (Gao et al., 2022), which can be challenging to obtain in practice. Further, the reconstructed dynamics recover slow, continuous motions (Pumarola et al., 2020; Li et al., 2022; Luiten et al., 2024; Wu et al., 2024; Wang et al., 2025) or color variations on quasi-static geometry (Lin et al., 2023). Another line of research integrates a continual learning framework, transforming the initial reconstruction to match gradual changes over time. The extension of 3DGS (Ackermann et al., 2025; Zeng et al., 2025) is inherently faster in rendering compared to NeRFs (Wu et al., 2023; Zhipeng Cai, 2023), and therefore enjoys faster training times. However, these works still require more than ten input images to adapt to scene changes and eventually lose information from previous time steps.

### 2.2 FEW-SHOT NERF AND GAUSSIAN SPLATTING

Several works pioneered relieving the dense-view requirement of NeRFs or 3DGS by incorporating geometric priors or regularization techniques (Niemeyer et al., 2022; Guangcong et al., 2023; Zhu et al., 2024; Uy et al., 2023). Recent geometric vision foundation models such as MASt3R (Leroy et al., 2024) provide strong geometric priors, enabling effective scene representation when combined with Gaussian splatting (Fan et al., 2024; Smart et al., 2024), particularly in cases where conventional structure-from-motion (SfM) (Schönberger & Frahm, 2016) fails. However, in the context of updating scene changes, these methods cannot preserve the initial reconstruction, which leads to significant performance degradation, such as severe floating artifacts in novel views. Our approach instead builds a geometric structure of reusable priors by aggregating actual observations, and sustains consistent long-term temporal reconstruction despite a sparse set of temporal image captures.

### 2.3 CHANGE DETECTION AND SEGMENTATIONS IN GAUSSIAN SPLATS

While change detection has remained a long-standing problem in computer vision (Sakurada et al., 2020; Sachdeva & Zisserman, 2022), vision foundation models facilitate more generalizable ways to detect changes. Recent approaches (Kim & Kim, 2025; Adam et al., 2023; Lu et al., 2025) estimate change regions by leveraging the encoded feature embeddings of the segment anything model (SAM) (Kirillov et al., 2023). Similarly, DINO features (Oquab et al., 2023) are leveraged to detect change regions (Ackermann et al., 2025; Galappaththige et al., 2025). Recently, several approaches have demonstrated that one can separate 3D objects from 3DGS with 2D masks (Shen et al., 2024; Zhang et al., 2025). Our framework incorporates a similar framework to build Gaussian templates of the shared objects in evolving 3D environments.

## 3 METHOD

### 3.1 OVERVIEW

Given the initial 3DGS reconstruction of the scene $\mathcal{G}_0$, our goal is to update the scene from a set of images $\mathcal{I} = \{I^i\}_t$, where $t \in \{t_1, t_2, \ldots, t_M\}$ represents a sparsely sampled time stamps, and the input images for each time stamp are captured from a small number of viewpoints $i = 1, \ldots, N_t$ that are not fixed beforehand. Our goal is to acquire the temporal evolution of the scene $\mathcal{S} = \{\mathcal{G}_0, \mathcal{G}_1, \ldots \mathcal{G}_M\}$ that recovers its states at the samples of time in the input. The representations are a

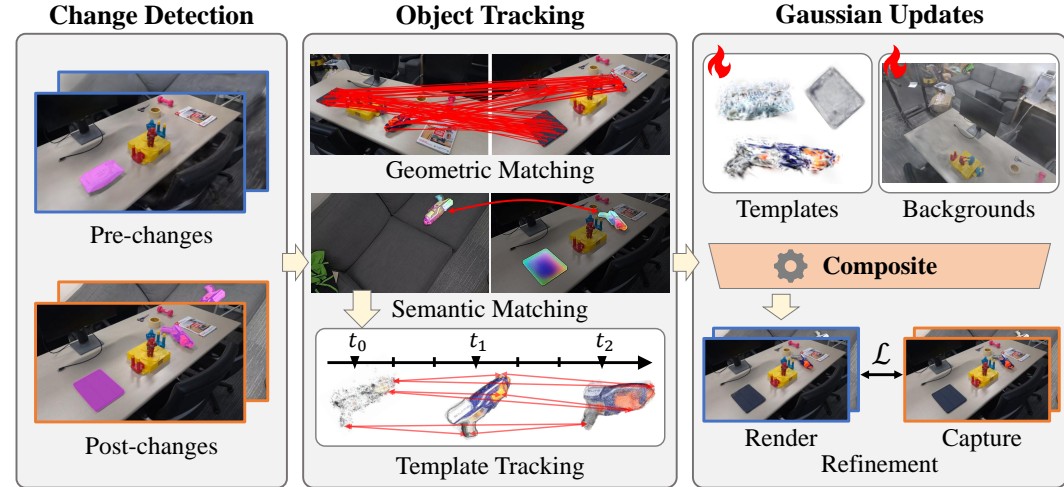

Figure 2: **Method overview.** We propose an integrated pipeline to update an initial reconstruction given the collection of post-change captures. Our pipeline first finds the camera locations of the input capture and compares them against the rendering of the initial reconstruction in the same view to detect object-level changes. We aggregate the detected objects in change from multiple viewpoints and different time stamps to create 3D Gaussian templates. We finally update the temporal scenes by compositing the object-level templates at their respective states with the background.

set of Gaussian splats, and each splat element is parameterized as $\{\mu, q, s, \alpha, c\}$, where $\mu \in \mathbb{R}^{N \times 3}$ denotes 3D center position, $q \in \mathbb{R}^{N \times 4}$ represents the quaternion representation of orientation (we denote its conversion into a rotation matrix as $R$), $s \in \mathbb{R}^{N \times 3}$ specifies scale, $\alpha \in \mathbb{R}^{N \times 1}$ denotes opacity, and $c \in \mathbb{R}^{N \times 48}$ encodes view-dependent color using third-order spherical harmonics.

Assuming the scene is an everyday environment with frequent interactions, it may experience geometric changes due to object displacements. We focus on modeling the object-level movements and develop a decomposed representation that can serve as a strong geometric prior despite sparse observations, as shown in Figure 2. Specifically, we robustly detect object-level changes (Section 3.2) and collect information for individual objects from the input image set (Section 3.3). We separately train Gaussian templates for the objects and static background (Section 3.4), such that blending them with the correct transform can best match the input images at the corresponding time step. Overall, our framework can achieve high-quality reconstruction in temporally evolving scenes while suppressing prominent artifacts with minimal input and computation.

## 3.2 CHANGE DETECTION

To decompose our scene representation into movable objects and static background, we detect changes from input images $\{I^i\}_t$. We first find the exact camera positions despite local changes, such that we can compare the observations from different time stamps against the initial Gaussian reconstruction $\mathcal{G}_0$. We use a robust hierarchical localization pipeline (Sarlin et al., 2019) and render the initial reconstruction in the same viewpoints $\{\hat{I}^i\}_t$.

We incorporate both semantic and photometric criteria to identify the differences between the rendered $\{\hat{I}^i\}_t$ and the captured $\{I^i\}_t$ images. Semantic differences detect object-level changes despite lighting variations and other adversaries, and are measured by the cosine similarity of SAM features (Kirillov et al., 2023), similar to recent change detection methods (Kim & Kim, 2025; Adam et al., 2023; Lu et al., 2025). Photometric differences are evaluated using the structural similarity index measure (SSIM), which can detect subtle object deviations that are not observable by SAM. The pixel-wise differences of the combined criteria are binarized by a scene-specific threshold chosen by the statistics (Otsu et al., 1975), resulting in an initial pseudo mask for changed regions.

We further refine the object boundaries of the binary image with the help of SAM masks. Specifically, we select the object region to be the set of SAM masks that sufficiently overlap with the

pseudo masks while containing semantically dissimilar features compared to the initial image, as the result of change (Kim & Kim, 2025). The aid of SAM masks effectively ignores differences due to floating artifacts in the rendering images and reliably extracts object regions. The resulting detected masks are dilated by 3 pixels to maintain sufficient information for 3D aggregation across multi-view observations, mitigating the impact of pixel-wise errors or slight misalignments.

### 3.3 OBJECT TRACKING AND TEMPLATE RECONSTRUCTION

After obtaining change masks from Section 3.2, we match the changed objects within the individual images and extract initial 3D Gaussian templates for them. We then refine the relative transforms between the 3D templates and the image observations, which can serve as input to integrated optimization for 3D scene $\mathcal{S}$.

**2D instance matching.** While one can associate object masks using low-level image features such as SIFT Lowe (2004), our sparse setting makes it challenging to match small objects with few discriminative features. Also, objects are sometimes dynamic in appearance or geometry, which further complicates the process. To address these issues, we combine the strength of both dense geometric features from MASt3R (Leroy et al., 2024) and semantic features obtained from SAM (Kirillov et al., 2023) in Section 3.2. We first match multi-view images incorporating MASt3R features (Leroy et al., 2024) for all pairs of images within the same time stamp $\{I^i\}_t$. Using the output, we build a graph where the nodes are object masks and the edges are matches with pairwise correspondences. By examining the graph structure, we can assign instance IDs to the matched components and filter out unmatched objects, such that we can naturally overcome artifacts. The intra-timestep matching then provides reliable starting points for establishing matchings across different timesteps. We aggregate the SAM features within the object region, which have been computed to detect changes in Section 3.2, and build a matrix that records extensive pairwise cosine similarity of the aggregated features. Based on the matrix, we leverage Hungarian matching (Munkres, 1957) to match object instances.

**Gaussian object template extraction.** After associating 2D change masks, we are ready to build object-level Gaussian templates, which can be refined and function as a reusable geometric prior. We first decompose the initial Gaussian reconstruction $\mathcal{G}_0$ into a set of template objects $\mathcal{T}_0 = \{o_{0,k}\}$ and the background $\mathcal{B}_0$. We formulate it as a segmentation problem for individual splats, and solve for the optimal label assignment as proposed in Shen et al. (2024). Objects in a later sequence do not exist in $\mathcal{G}_0$, and we initialize the templates $\{\mathcal{T}_t | t > 0\}$ using 3D point clouds estimated with MASt3R. The point clouds directly provide the positions and colors of the Gaussian splats, and we simply initialize uniform opacity, identity rotation, and uniform scaling as Fan et al. (2024). For the background, we maintain a single global set of Gaussians $\mathcal{B}_0$. When occluded regions in the initial reconstruction become visible after changes, we augment this background representation by initializing the newly observed areas using point maps from MASt3R.

**Gaussian object template tracking.** After initializing 3D templates, we deduce temporal states of the objects by tracking their movements and verifying consistency in 3D. For the object instances matched from different time steps, we compare their 3D overlap using robust point cloud registration. However, the MASt3R points or Gaussian reconstructions are incomplete and noisy with irregular density, and often cannot be registered using conventional point cloud pipelines such as ICP (Besl & McKay, 1992) or RANSAC-based approaches (Fischler & Bolles, 1981). Instead, we establish correspondences by augmenting DINO features (Oquab et al., 2023) to each point and apply a robust point cloud registration pipeline (Yang et al., 2020) to register templates. The registration yields 6DoF poses between pairs of template points $P_{t \to \tilde{t}, k} = \{R_{t \to \tilde{t}, k}, T_{t \to \tilde{t}, k}\}$ such that we can assess the geometric consistency across the temporal track by thresholding with the Chamfer distance (Fan et al., 2017). If the points are close enough, we avoid redundancy by selecting a single 3D template per matched instance, along with its relative transforms over time. Since we conservatively select templates, we can naturally represent an object under significant geometric variations as different instances without modifying the framework, as shown in Figure 6. We further refine the shared template in the next stage.

### 3.4 LONG-TERM GAUSSIAN SPLATS OPTIMIZATION

While the selected templates can explain the other time steps with sufficient geometric overlap, they are only derived from a single time step and can be noisy and incomplete, especially in different

time steps with significant viewpoint changes or other transforms. We aggregate the collection of observations and optimize the parameters of Gaussian splats (Section 3.1). The templates from a time step $t$ can be transformed into a different time step $\tilde{t}$ using the registration parameters $P_{t \to \tilde{t}, k} = \{R_{t \to \tilde{t}, k}, T_{t \to \tilde{t}, k}\}$ from Section 3.3 as following:

$$(\mu_{\tilde{t},k}, R_{\tilde{t},k}, c_{\tilde{t},k}) = (\mu_{t,k} R_{t \to \tilde{t},k}^\top + T_{t \to \tilde{t},k}^\top, R_{t \to \tilde{t},k} R_{t,k}, c_{t,k} \mathcal{R}_{\text{SH}}(R_{t \to \tilde{t},k})^\top). \tag{1}$$

Here, $R_{t,k}$ and $R_{\tilde{t},k}$ are rotation matrices corresponding to $q_{t,k}$ and $q_{\tilde{t},k}$ respectively and SH coefficients are rotated via rotation operator $\mathcal{R}_{\text{SH}}$ (Chang et al., 2024). In addition, we apply a temporal opacity filter per object $\mathcal{M}_{t,o}$ such that transient objects become invisible (zero opacity). Since the 6DoF poses obtained from Section 3.3 are not precise at the pixel level, we additionally set the 6DoF poses of the object templates as an optimization parameter.

These strategies enable efficient modeling of long-term environments, but they also risk overfitting the initial assets to post-change views. To address this issue, we additionally leverage training camera poses used at the initial stages to render the scene and enforce consistency in the rendered images. Formally, we can render and update the image from the $i$th viewpoint at time $t$ by defining the optimization problem as follows:

$$(\mu_{t,k}, R_{t,k}, c_{t,k}) = (\mu_{0,k} R_{0 \to t,k}^\top + T_{0 \to t,k}^\top, R_{0 \to t,k} R_{0,k}, c_{0,k} \mathcal{R}_{\text{SH}}(R_{0 \to t,k})^\top),$$

$$\hat{I}_t^i = \text{Rasterize}(\mu_{t,k}, q_{t,k}, s_{t,k}, \mathcal{M}_{t,o} \cdot \alpha_{t,k}, c_{t,k}), \tag{2}$$

$$\{\mu^*, q^*, s^*, \alpha^*, c^*, P^*\} = \arg\min \mathcal{L}_{\text{photo}}\left(\hat{I}_t^i, I_t^i\right).$$

where $P = \{P_{0 \to t,k} \mid \forall t, k\}$ and $I_t^i$ contains both captured post-change images and renderings from the initial camera poses. For $\mathcal{L}$, we use the standard L1 loss with D-SSIM loss that was used in the original implementation of 3DGS (Kerbl et al., 2023). The background scene is initialized with $\mathcal{B}_0$ and similarly refined using all the information from different times. As the initial templates provide a reasonable approximation, 5000 iterations are sufficient to refine the parameters without densifying or cloning Gaussians, and we also skip the opacity resetting stages. Once optimized, our framework easily scales to multiple timesteps by simple transformations of template Gaussians.

## 4 EXPERIMENTS

### 4.1 DATASETS & BASELINES

**Datasets.** We use a synthetic dataset from CL-NeRF Wu et al. (2023), which contains three scenes captured from different timesteps: [WHITEROOM, KITCHEN, ROME]. Each timestep includes object-level sequential operations, including addition, deletion, replacement, and movement. However, the motions are simple and do not exhibit diverse variations between objects in different steps. We additionally captured real-world scenes with more challenging settings, where objects may abruptly reappear in different configurations. The dataset consists of image collections captured in five scenes at 5 different timesteps: [CAFE, DININGROOM, HALL, LAB, LIVINGROOM].

**Baselines.** We compare our work to recent NeRFs and 3DGS variants. First, we evaluate against the original (1) 3DGS (Kerbl et al., 2023) using all images for different timesteps as a reference. We further compare with (2) InstantSplat (Fan et al., 2024), a few-shot reconstruction method applied independently at each timestep. To account for frameworks explicitly modeling dynamic scenes, we include (3) 4DGS (Wu et al., 2024) and (4) Neural Scene Chronology (NSC) (Lin et al., 2023). In addition, we compare (5) 3DGS-CD, which explicitly detects and updates object-level changes. Finally, we evaluate against continual learning frameworks, including (6) CL-NeRF (Wu et al., 2023) and (7) CL-Splats (Ackermann et al., 2025). The works and their capabilities are also summarized in Table 1.

### 4.2 COMPARATIVE STUDIES

We evaluate our method and baselines on novel-view synthesis tasks across scenes with multiple timesteps. For every setting, we use three images at each timestep from various angles to capture scene changes. Our framework largely outperforms baselines both qualitatively and quantitatively, as demonstrated in Table 2 and Figure 3. It successfully reconstructs diverse object-level changes

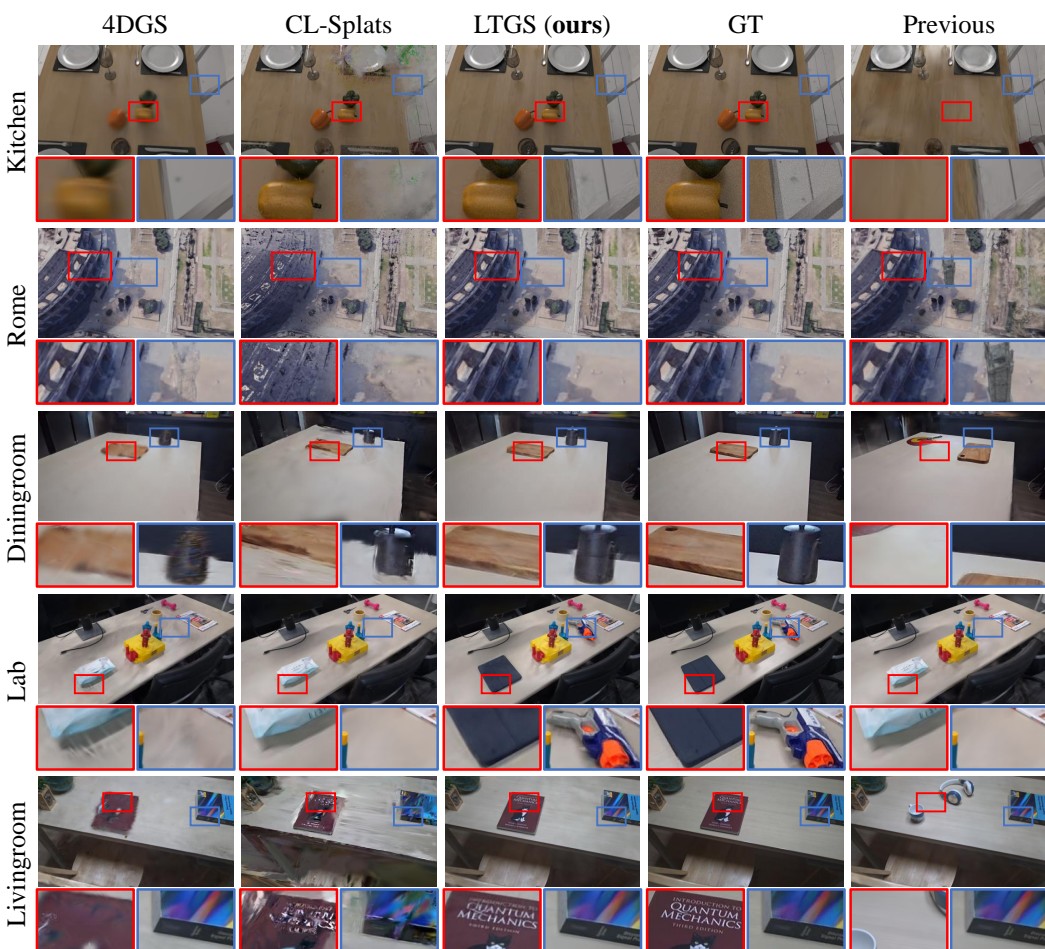

Figure 3: **Qualitative comparisons of our method.** We illustrate the results of our method using the CL-NeRF dataset and our dataset.

Table 2: **Quantitative comparisons on CL-NeRF dataset and our dataset.** We compared our work to the variants of NeRFs and Gaussian splatting.

| Method ↓ | CL-NeRF dataset (synthetic) | | | Our dataset (real) | | | Time |
|---|---|---|---|---|---|---|---|
| | PSNR ↑ | SSIM ↑ | LPIPS ↓ | PSNR ↑ | SSIM ↑ | LPIPS ↓ | |
| 3DGS (Kerbl et al., 2023) | 24.53 | 0.789 | 0.392 | 19.56 | 0.857 | 0.272 | 7 min |
| InstantSplat (Fan et al., 2024) | 18.98 | 0.601 | 0.466 | 19.36 | 0.785 | 0.343 | 3 min |
| 4DGS (Wu et al., 2024) | 26.13 | 0.786 | 0.411 | 21.49 | 0.850 | 0.322 | 27 min |
| NSC (Lin et al., 2023) | 20.63 | 0.698 | 0.465 | 17.52 | 0.755 | 0.439 | 20 hours |
| 3DGS-CD (Lu et al., 2025) | 23.61 | 0.727 | 0.437 | 20.94 | 0.774 | 0.348 | 2 min |
| CL-NeRF (Wu et al., 2023) | 25.53 | 0.730 | 0.465 | 20.95 | 0.815 | 0.379 | 2 hour |
| CL-Splats (Ackermann et al., 2025) | 25.84 | 0.772 | 0.416 | 21.12 | 0.829 | 0.312 | 3 min |
| LTGS (ours) | **27.17** | **0.795** | **0.376** | **23.46** | **0.889** | **0.230** | 6 min |

while remaining robust to limitations imposed by sparse views. In particular, InstantSplat (Fan et al., 2024) is designed for fast and lightweight reconstruction specifically in a few-shot setting, and thus cannot maintain its performance on a free-viewpoint setting covering the full scene. 4DGS (Wu et al., 2024) and NSC (Lin et al., 2023) struggle to precisely model the discrete changes, such as added or removed objects. The snapshots in Figure 3 also include overly smooth results in the change regions.

While 3DGS-CD (Lu et al., 2025) also quickly handles object-level placement changes, our approach consistently outperforms, as our approach better accounts for added and removed objects.

Table 3: **Ablation studies on different optimization configurations.**

| Configuration | PSNR ↑ | SSIM ↑ | LPIPS ↓ |
|---|---|---|---|
| w/o Obj. Tracking | 23.26 | 0.885 | 0.234 |
| w/o Pose Opt. | 23.33 | 0.886 | 0.232 |
| w/o BG Init. | 23.29 | 0.885 | 0.233 |
| w/o Training View | 23.11 | 0.885 | 0.240 |
| Full (ours) | **23.46** | **0.889** | **0.230** |

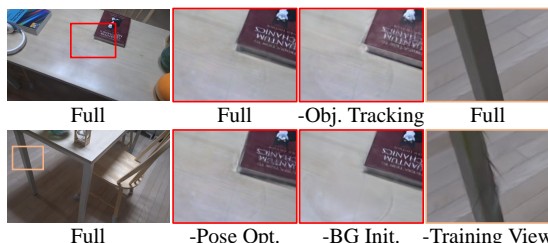

Figure 4: **Visual comparison of ablation study.**

Recent continual-learning frameworks also struggle to address spatio-temporally sparse settings. CL-NeRF (Wu et al., 2023) performs well on synthetic datasets but cannot track complex real-world changes. Also, the implicit representation of CL-NeRF occasionally results in degraded sharpness, as shown in the appendix. The optimization of CL-Splats (Ackermann et al., 2025) fails to maintain its stability in the sparse-view inputs, as its mask estimation becomes unreliable. The effects can also be observed in the LAB scene (4th row) in Figure 3.

We also report the total time spent reconstructing different timesteps in Table 2. In our framework, the total processing time is approximately 6 minutes (2.5 min for change detection, 0.5 min for instance matching, 3 min for 5000 iteration updates), measured on an NVIDIA RTX 4090. In conclusion, while prior methods either oversmooth dynamic changes or fail under sparse inputs, our framework achieves accurate and efficient reconstruction of evolving scenes with strong object-level consistency.

### 4.3 ABLATION STUDIES

We conducted ablation studies on various components of the method, as demonstrated in Table 3. Since our components are primarily designed to improve the modeling of changing objects rather than the overall quality of the initially reconstructed scenes, the image quality metric itself did not fully reveal significant improvement. Accordingly, we additionally demonstrate the qualitative comparisons focusing on the change regions in Figure 4.

We first tested the effect of instance matching, where we removed the template association step and built new object-level Gaussians at every time step. Without using object Gaussian templates as a reusable prior, we could not handle sparse view limitations, leaving traces of removed objects as shown in Figure 4. 6DoF pose updates also increased the reconstruction quality, as they account for pixel-level errors from subtle pose errors after registration.

While our primary ablations focus on reconstructing dynamic objects, we also examined the impact of background initialization. Specifically, when objects disappear and leave previously occluded regions visible, initializing these empty areas with MASt3R (Leroy et al., 2024) point clouds alleviates background artifacts. As shown in Figure 4, the joint use of global background Gaussians with this initialization strategy further reduces artifacts after object removal. Including training views of initial timesteps also enhanced the quality, as reported in Table 3. As we only leverage a few-shot images, some unseen regions, such as under the table or back of the chair in Figure 4, include sharp artifacts, which degrade the rendering of several viewpoints. We verified that each component clearly affected the enhancement of both object and background reconstructions without compromising the initial reconstruction.

### 4.4 APPLICATIONS

**Object-level reconstruction.** After optimization, our framework produces object-level reconstruction, where the learned Gaussian templates can be directly leveraged for scene composition and temporal reasoning. To further illustrate this, we visualize the optimized object-level Gaussian templates in Figure 5. We selected some viewpoints from initial and post-change captures, and rasterized the object-level Gaussians to the corresponding viewpoints. We demonstrate that our framework is capable of disentangling individual objects from the scene while preserving consistent geometry and appearance across different time steps. Notably, even with few-shot observations, the optimized object templates exhibit well-defined shapes without severe artifacts along object boundaries. This suggests that the optimization effectively integrates multi-timestep cues into coherent object-level

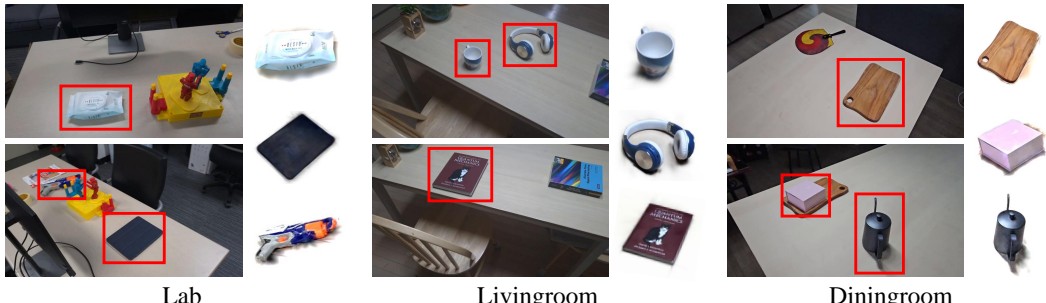

Lab      Livingroom      Diningroom

Figure 5: **Object template visualization.** We sampled several captures from the initial state and post-change captures and corresponding object-level Gaussian templates.

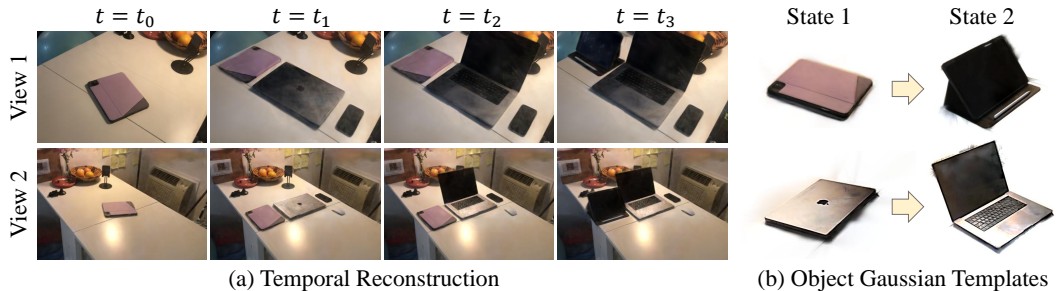

(a) Temporal Reconstruction      (b) Object Gaussian Templates

Figure 6: **Challenging real-world scenarios.** We demonstrate (a) reconstruction results with articulations and (b) a visualization of object-level reconstructions.

representations. Such clean object templates also highlight the potential of our method for modeling object-level changes in scenes with longer temporal variances.

**Non-rigid transformations or articulations.** We further tested our framework on more challenging setups as demonstrated in Figure 6. In real-world scenarios, there exist more challenging object-level changes, such as non-rigid transformations or articulations. Our framework fails to stably track dramatic geometric changes, as we can only track and build templates that satisfy both 2D mask matching and 3D alignments. Still, in those cases, we can define separate object Gaussian templates for those in different states. We show the reconstructed scene with temporal variation, and tracked object-level reconstruction with different states for the MAC scene from the world-across-time (WAT) dataset introduced in CLNeRF (Zhipeng Cai, 2023). Although our current formulation cannot fully reuse priors for such complex geometric changes, it provides a principled way to represent objects in different states through independent templates. We may extend it toward modeling richer geometric deformations and dynamic interactions in real-world environments in future work.

## 5 CONCLUSION

We present LTGS, an integrated framework for modeling scenes with long-term changes given spatiotemporally sparse images. Our strategy stably builds and exploits object-centric templates under the challenging setup. Several comparative studies and ablation studies have verified that the combination of our components significantly outperforms the baselines. We further verified our frameworks to be applicable to several extensions, such as object-level reconstruction and reconstructing more challenging setups with non-rigid transformations or articulations. In conclusion, we believe this framework offers a promising foundation for building a coherent structural representation that is reusable for a long temporal horizon. As our framework primarily targets scenes with geometric variations, it poses challenges for scenes with significant lighting changes or severe appearance changes with fixed geometries, such as monitors, if these are not captured as changes. We leave the problem of modeling the heavy lighting changes with shadows for future work.

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

# A  IMPLEMENTATION DETAILS

## A.1  CHANGE DETECTION

We detect fine object-level changes in 2D by using the semantic prior of the SAM model (Kirillov et al., 2023). We prepare a pair of images $(I_t^i, \hat{I}_t^i)$ from captured and rendered images from the corresponding viewpoints. To find abstract differences between the rendered and captured images, we extract features from the segmentation model for both images. We used features obtained from the pretrained SAM encoder and interpolated them to match the original image resolution, after which we computed pairwise cosine similarities for comparison. We additionally use SSIM to capture structural differences, where using only semantic cosine similarity struggles to find slightly deviated objects.

To obtain coarse change masks, we need a binarization by thresholding the obtained differences. Since the binary coarse mask is highly sensitive to manually defined thresholds, which vary significantly across scenes, we adopt Otsu's method (Otsu et al., 1975) to automatically determine the threshold $\tau_{cos}$. We obtain the coarse binary masks $\mathcal{M}_{t,\text{coarse}}^i$ as follows:

$$M_{t,\text{coarse}}^i = \gamma \cdot \cos(\mathcal{E}(I_t^i), \mathcal{E}(\hat{I}_t^i)) + (1 - \gamma) \cdot \text{SSIM}(I_t^i, \hat{I}_t^i) \leq \tau_{\text{cos}}, \quad (3)$$

where $\mathcal{E}$ denotes the feature extractor of SAM.

After obtaining coarse change regions, we extract fine-grained object-level change masks to model instance-wise changes. Since precise object-level changes are difficult to capture using the coarse stages described above, we leverage the automatic mask generation from SAM Kirillov et al. (2023) within the coarse binary mask $M_{t,\text{coarse}}^i$. For each generated mask, we first calculate the intersection-over-union (IoU) between its region and the coarse binary mask. We further compare the cosine similarity between the accumulated features within those regions in the extracted features of the rendered and captured images. The fine object-level masks can be obtained as follows:

$$\mathcal{O}_t^i = \bigcup_k \left\{ o_{t,k}^i \,\middle|\, \text{IoU}(o_{t,k}^i, M_{t,\text{coarse}}^i) \geq \tau_{\text{IoU}} \wedge \cos\big(\Phi(o_{t,k}^i, \mathcal{E}(I_t^i)), \Phi(o_{t,k}^i, \mathcal{E}(\hat{I}_t^i))\big) \leq \tau_{\text{cos}} \right\}, \quad (4)$$

where $o_{t,k}^i$ denotes the $k$th object mask generated by automatic mask generation, and $\Phi(m, X) = \frac{1}{|m|} \sum_{p \in \Omega} m(p) \, X(p)$ denotes the average pooling operator of feature $X$ within the interior $\Omega$ of mask $m$. We select and keep only the highly overlapped and semantically different masks following Kim & Kim (2025). We further removed masks occupying only small regions to prevent noise, and we dilated the change masks to address pixel-wise errors. For change detection, we set $\gamma = 0.7$ and $\tau_{cos} = 0.9$ in Eq. 3 and Eq.4, which were used throughout our experiments.

## A.2  OBJECT TRACKING

We provide additional details of our pipeline to associate with changed instances. Given 2D object-level change masks from Section 3.2, we leverage both visual feature and SAM feature matching to associate 2D change masks. As mentioned in Section 3.3, we separate the instance matching into intra-timestep matching and cross-timestep matching. Let $\mathcal{O}_t^i = \{o_{t,1}^i, o_{t,2}^i, \ldots, o_{t,N_o}^i\}$ be the set of object-level masks of $i$th viewpoint at timestep $t$. We first compute pairwise matches across images $I_t^i$ and $I_t^j$ using MASt3R within the same timestep. Given the extracted MASt3R descriptors $\mathbf{d}_{t,k}^i$ and $\mathbf{d}_{t,l}^j$ for object $o_{t,k}^i$ and $o_{t,l}^j$ respectively, we find descriptor matches $\mathcal{M}_t^{(i,k)\leftrightarrow(j,l)}$ within object change masks as follows:

$$\mathcal{M}_t^{(i,k)\leftrightarrow(j,l)} = \text{match}(\mathbf{d}_{t,k}^i, \mathbf{d}_{t,l}^j), \quad o_{t,k}^i \in \mathcal{O}_t^i, \ o_{t,l}^j \in \mathcal{O}_t^j, \ i \neq j. \quad (5)$$

Note for matching, we follow the fast reciprocal matching procedure from MASt3R Leroy et al. (2024). Based on these matches, we construct a graph $G_t = (N_t, E_t)$ as:

$$N_t = \bigcup_i \mathcal{O}_t^i, \quad E_t = \left\{ \big(o_{t,k}^i, o_{t,l}^j, |\mathcal{M}_t^{(i,k)\leftrightarrow(j,l)}|\big) \,\middle|\, o_{t,k}^i \in \mathcal{O}_t^i, \ o_{t,l}^j \in \mathcal{O}_t^j, \ i \neq j \right\}, \quad (6)$$

where the nodes $N_t$ are objects and the edge weight $E_t$ encodes the total number of matches between every pair of objects $o_{t,k}^i$ and $o_{t,l}^j$. We then cluster the graph using the depth-first search (DFS)

algorithm (Tarjan, 1972) to identify connected components, where each component corresponds to a unique global object identity. Based on these clustering results, we assign instance IDs to the matched components and filter out unmatched objects for consistency. This filtering strategy gives robustness to detected instances that are inaccurate due to the artifacts in rendered images.

After obtaining object-level matches for every sequence within an identical timestep, it is essential to associate object masks across different timesteps. For each object $o_{t,k}^i$ and $o_{\tilde{t},l}^j$ where $t$ and $\tilde{t}$ are the set of target times after intra-timestep matching, we accumulate the SAM (Kirillov et al., 2023) features in the object region, and build a matrix $\mathbf{S}_{k\leftrightarrow l}$ that contains cosine similarity among every pair as follows:

$$\mathbf{S}_{k\leftrightarrow l} = \cos\big(\Phi(o_{t,k}^i, \mathcal{E}(I_t^i)), \Phi(o_{\tilde{t},l}^j, \mathcal{E}(I_{\tilde{t}}^j))\big). \tag{7}$$

Based on the matrix, we leverage Hungarian matching (Munkres, 1957) to solve an optimal assignment problem between the instances as $\pi^* = \arg\max_\pi \sum_k \mathbf{S}_{k\leftrightarrow\pi(k)}$, where $\pi(k)$ denotes the matched object in timestep $\tilde{t}$. Note that we conduct this process identically for every possible timestep pair for $t \in [0, T]$.

### A.3 OBJECT GAUSSIAN TEMPLATE RECONSTRUCTION

Given the tracked object masks $M = \{M_t^i | i = 1, ..., N_v; t = 0, ..., T\}$, we provide additional details of the construction and initialization of object-level Gaussian Splats. Here, we define the total number of instances at the initial timestep as $E$. For the objects that emerge in initial reconstruction, we separate those using the optimal label assignment problem introduced in FlashSplat (Shen et al., 2024). For our task, the problem is defined as follows:

$$\min_{\{P_k\}} \mathcal{F} = \sum_i \left| \sum_k P_k \alpha_k T_k - M_0^i \right|, \quad P_k \in \{0, 1, ..., E\}, \tag{8}$$

where $\alpha_k, T_k$ each denotes the alpha value and transmittance during volume rendering, and $P_k$ denotes the per-Gaussian 3D label. Among the $P_k$, index 0 corresponds to the background, while the remaining indices correspond to the foreground. The above equation solves the problem of assigning the 3D label $P_k$ by volume rendering them to the image domain to match the given multiview masks at $t = 0$. Specifically, we use the majority voting algorithm as follows:

$$\begin{aligned}
P_k &= \arg \max_{n \in \{0, m\}} A_n, \\
A_m &= \sum_i \alpha_k T_k \mathbb{1}(M_0^i, m), \\
A_0 &= \sum_i \sum_{e \neq m} \alpha_k T_k \mathbb{1}(M_0^i, e),
\end{aligned} \tag{9}$$

where $\mathbb{1}(M_0^i, m)$ denotes the indicator function which is equal to 1 if the pixel in mask $M_0^i$ belongs to object $m$, and 0 otherwise. Eq. 9 solves the assignment problem by allocating the label that maximizes the weighted contribution of Gaussians within the object mask regions. Please refer to the original FlashSplat (Shen et al., 2024) paper regarding the details and the derivation. We additionally filter Gaussians that are out of the object mask region after directly projecting the centers to remove floating artifacts.

After registration and geometric verification as presented in Section 3.3, we initialize Gaussians for objects that do not exist in the initial reconstruction. We first extract point clouds for new objects from the global scene reconstruction of MASt3R (Leroy et al., 2024) with estimated poses from the hierarchical localization pipeline Sarlin et al. (2019). In the original implementation of MASt3R, camera parameters were optimized jointly with per-view depth maps and global scales. We modify the optimization loop to operate only on depth maps with scale and offset parameters, while the camera poses remain fixed. To reduce noise, we retain only the point clouds with per-pixel confidence values greater than 1.5, and we randomly downsample the point cloud by a factor of 4, as per-pixel point clouds are overly dense, which is inefficient for optimization.

To initialize Gaussian primitives, we use the dense point cloud's position and its corresponding color as the initial position and color. We convert the RGB color into spherical harmonics (SH)

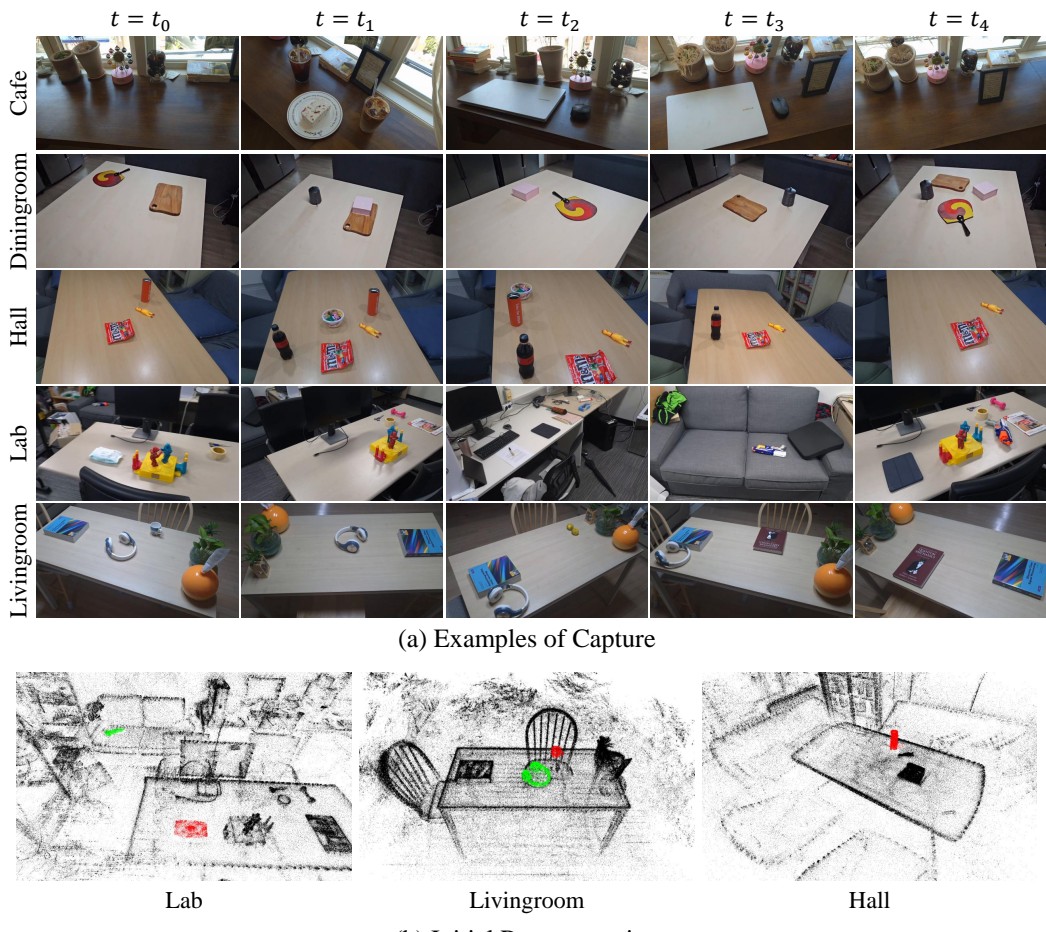

(a) Examples of Capture

Lab Livingroom Hall

(b) Initial Reconstruction

Figure 7: **Examples of our datasets.** (a) Illustration of long-term captures of our dataset for each scene. (b) Initial reconstruction of Gaussian splats at $t = 0$ and tracked instances at the initial timestep.

coefficients and initialize higher-order SH components with zeros. We use low initial opacity values as $\alpha = 0.1$ for all Gaussians. Rotations are initialized as identity quaternions, i.e., $q = (1, 0, 0, 0)$ for all Gaussians, while the scales are determined by the pairwise squared distance as done in the original 3DGS (Kerbl et al., 2023).

## B  DATASETS

In this section, we provide some additional details for the datasets that we have introduced in our main manuscript. We casually captured video sequences using Galaxy S24 with fixed focal lengths and manually changed several objects between the captures for 5 sequences. After converting video frames to images, we regularly sampled 300-400 images for initial reconstruction to cover the scene of interest, and obtain camera poses using COLMAP SfM (Schönberger & Frahm, 2016). Figure 7 illustrates the example of our datasets for every scene, captured from different timesteps. In total, our dataset contains five scenes that cover a diverse set of indoor spaces (Cafe, Diningroom, Livingroom, Hall, Lab). We additionally visualized the initial reconstruction of sampled scenes, with tracked instances using our pipeline, which serve as the initial object Gaussian templates before refinement. For evaluation, we selected every 8th frame following the conventional evaluation protocol of neural rendering (Mildenhall et al., 2019; 2020).

## C ADDITIONAL RESULTS

### C.1 ADDITIONAL QUALITATIVE COMPARISONS

We present additional qualitative results for both CL-NeRF Wu et al. (2023) and our datasets in Figure 8. We illustrated scenes that are not covered in Figure 3 of the main manuscript. By incorporating reusable priors into scene representations, we achieve a notable reduction of artifacts, particularly in under-constrained regions where other baselines often struggle. Compared to the baselines, our method produces cleaner geometry and more photorealistic synthesis under novel viewpoints. This improvement highlights the effectiveness of our method in reducing ambiguity and enabling stable reconstructions across diverse scenes.

### C.2 PER-SCENE QUANTITATIVE COMPARISONS

We provide the evaluation results of all scenes in terms of PSNR, SSIM, and LPIPS. As demonstrated in Tables 4, 5 and 6, our framework has achieved the best results for most scenes.

Table 4: **PSNR comparisons on CL-NeRF dataset and our dataset.** The first and second best results are highlighted in **bold** and underlined, respectively.

| Method | CL-NeRF dataset | | | Our dataset | | | | |
| --- | --- | --- | --- | --- | --- | --- | --- | --- |
| | Whiteroom | Kitchen | Rome | Cafe | Diningroom | Livingroom | Hall | Lab |
| 3DGS (Kerbl et al., 2023) | 20.66 | 24.55 | 28.38 | 15.97 | 18.44 | 20.44 | 22.35 | 20.60 |
| InstantSplat (Fan et al., 2024) | 19.35 | 17.66 | 19.92 | 17.71 | 22.14 | 19.30 | 18.66 | 18.98 |
| 4DGS (Wu et al., 2024) | 24.84 | 25.41 | 28.15 | 18.01 | 22.84 | 22.51 | 23.33 | 20.76 |
| NSC (Lin et al., 2023) | 18.13 | 17.40 | 26.36 | 15.10 | 18.21 | 19.81 | 18.63 | 15.86 |
| 3DGS-CD (Lu et al., 2025) | 23.86 | 22.20 | 24.77 | 17.94 | 22.15 | 21.00 | 22.22 | 21.41 |
| CL-NeRF (Wu et al., 2023) | 26.22 | 25.81 | 24.55 | 16.53 | 22.66 | 22.32 | 22.40 | 20.87 |
| CL-Splats (Ackermann et al., 2025) | **26.84** | 24.90 | 25.79 | 18.71 | 19.29 | **23.94** | 22.21 | 21.47 |
| LTGS (ours) | 26.67 | **26.21** | **28.65** | **20.77** | **25.22** | 23.50 | **25.22** | **22.64** |

Table 5: **SSIM comparisons on CL-NeRF dataset and our dataset.** The first and second best results are highlighted in **bold** and underlined, respectively.

| Method | CL-NeRF dataset | | | Our dataset | | | | |
| --- | --- | --- | --- | --- | --- | --- | --- | --- |
| | Whiteroom | Kitchen | Rome | Cafe | Diningroom | Livingroom | Hall | Lab |
| 3DGS (Kerbl et al., 2023) | 0.821 | **0.645** | **0.899** | 0.776 | 0.866 | 0.897 | 0.911 | 0.835 |
| InstantSplat (Fan et al., 2024) | 0.699 | 0.443 | 0.660 | 0.723 | 0.857 | 0.795 | 0.809 | 0.739 |
| 4DGS (Wu et al., 2024) | 0.827 | 0.644 | 0.885 | 0.772 | 0.893 | 0.882 | 0.892 | 0.812 |
| NSC (Lin et al., 2023) | 0.710 | 0.536 | 0.849 | 0.705 | 0.795 | 0.813 | 0.804 | 0.658 |
| 3DGS-CD (Lu et al., 2025) | 0.815 | 0.563 | 0.803 | 0.719 | 0.792 | 0.797 | 0.751 | 0.812 |
| CL-NeRF (Wu et al., 2023) | 0.829 | 0.636 | 0.725 | 0.696 | 0.863 | 0.881 | 0.859 | 0.775 |
| CL-Splats (Ackermann et al., 2025) | **0.848** | 0.627 | 0.840 | 0.749 | 0.873 | 0.836 | 0.866 | 0.819 |
| LTGS (ours) | **0.848** | **0.645** | 0.892 | **0.845** | **0.911** | **0.922** | **0.924** | **0.840** |

Table 6: **LPIPS comparisons on CL-NeRF dataset and our dataset.** The first and second best results are highlighted in **bold** and underlined, respectively.

| Method | CL-NeRF dataset | | | Our dataset | | | | |
| --- | --- | --- | --- | --- | --- | --- | --- | --- |
| | Whiteroom | Kitchen | Rome | Cafe | Diningroom | Livingroom | Hall | Lab |
| 3DGS (Kerbl et al., 2023) | 0.522 | 0.537 | **0.116** | 0.323 | 0.313 | 0.216 | 0.234 | 0.273 |
| InstantSplat (Fan et al., 2024) | 0.561 | 0.579 | 0.257 | 0.328 | 0.295 | 0.367 | 0.377 | 0.346 |
| 4DGS (Wu et al., 2024) | 0.539 | 0.547 | 0.148 | 0.363 | 0.306 | 0.307 | 0.299 | 0.334 |
| NSC (Lin et al., 2023) | 0.585 | 0.619 | 0.192 | 0.441 | 0.385 | 0.426 | 0.431 | 0.510 |
| 3DGS-CD (Lu et al., 2025) | 0.527 | 0.571 | 0.213 | 0.337 | 0.366 | 0.345 | 0.375 | 0.318 |
| CL-NeRF (Wu et al., 2023) | 0.521 | 0.536 | 0.339 | 0.463 | 0.324 | 0.328 | 0.350 | 0.428 |
| CL-Splats (Ackermann et al., 2025) | 0.492 | 0.542 | 0.214 | 0.345 | 0.308 | 0.323 | 0.303 | 0.280 |
| LTGS (ours) | **0.478** | **0.522** | 0.128 | **0.246** | **0.253** | **0.185** | **0.204** | **0.260** |

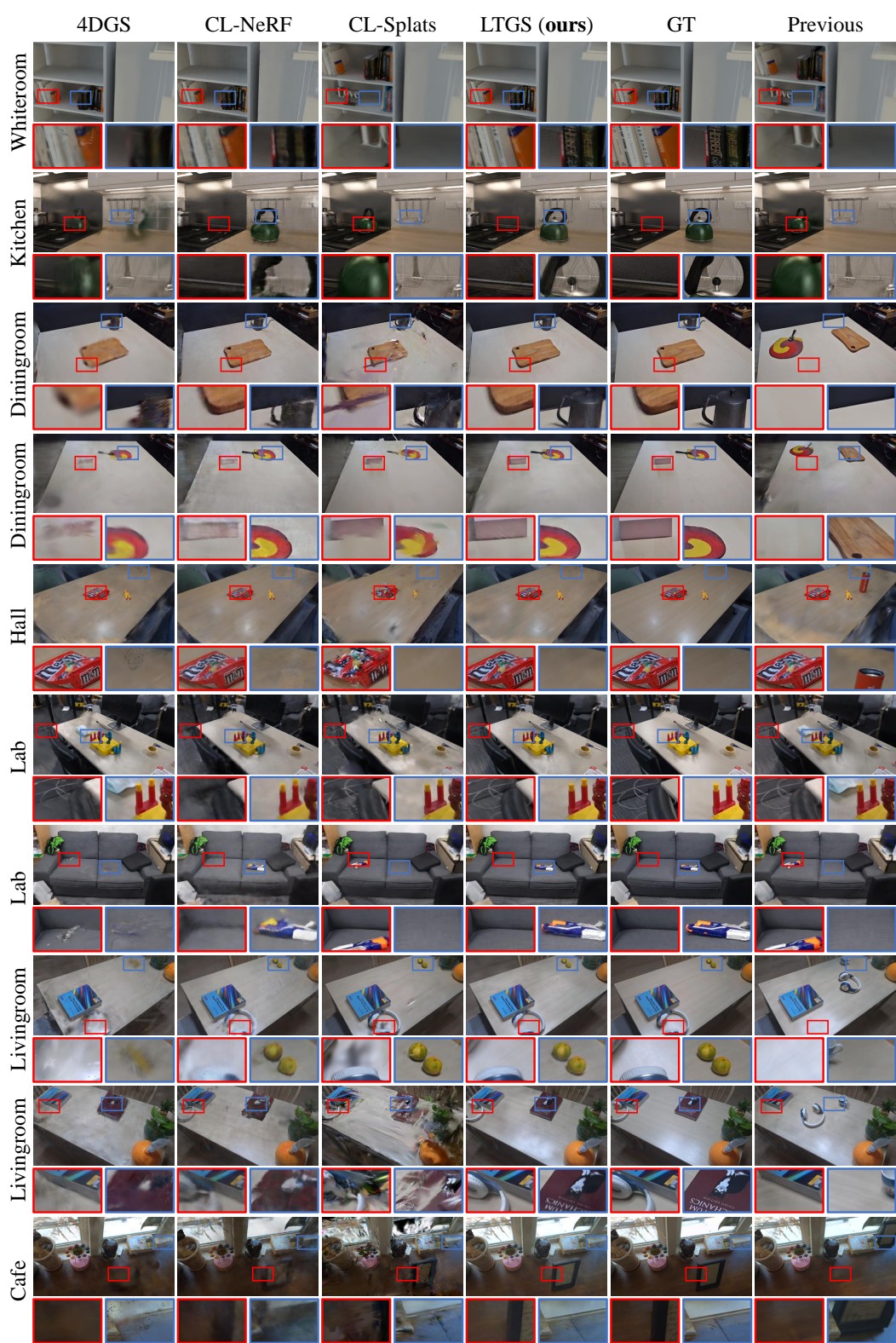

Figure 8: **Additional qualitative comparisons.** We illustrate the results of our method and baselines using CL-NeRF dataset and our dataset.

