# OpenReview forum: "LTGS: Long-Term Gaussian Scene Chronology From Sparse View Updates"
_ICLR.cc/2026/Conference — ICLR 2026 Conference Withdrawn Submission_

### Official Review · Reviewer_3N9D · 2025-10-28

**Soundness:** 2
**Presentation:** 3
**Contribution:** 2
**Rating:** 4
**Confidence:** 3

**Summary:**

This work tackle the task for updating the reconstructed 3DGS, where some of the objects are moved, added, or deleted, from few images captured in sparse time steps. To this end, this work try to separate the scene into dynamic objects and background part so different time step can be explained by blending the transformed dynamic objects and the background. The entire pipeline has several stages to reconstruct the dynamic object templates: dynamic objects segmentation, instance matching, tracking, and 3DGS optimization using the initial and the updated views. The results show significant improvement comparing to previous dynamic 3DGS methods.

**Strengths:**

The proposed new setup is interesting and could be useful in practice. If we use 3DGS as map, users or robot navigators may want the ability to quickly updating the map from few observations.

**Weaknesses:**

My main concerns is about its long pipeline and somewhat ad-hoc approach. Each stage in the pipeline is heuristic algorithm aided by several foundation models (e.g., SAM, DINOv2) for change detection, instance matching and tracking. These are all very important and fundamental task in CV and will be very nice if the proposed algorithm can improve them. However, these components don't seems will validated in this work. The results are showcased on eight scenes where five of them are newly collected by this work. As the proposed method are a sequence of heuristics algorithms, I'm not so convinced regarding the robustness. Besides, I think the hyperparameters and their effect need more analysis which will gives users hint on how to tune them in case some of the stages fail.

**Questions:**

In case the background or objects are occluded or only partially observed at initial, how does the proposed method can recover or complete them? It seems that the proposed method does not have a way to add more Gaussians.

---

### Official Review · Reviewer_PKSH · 2025-10-30

**Soundness:** 3
**Presentation:** 2
**Contribution:** 3
**Rating:** 4
**Confidence:** 3

**Summary:**

LTGS is designed to capture highly unconstrained, casually recorded variations in everyday scenes and to robustly model the long-term evolution of a scene. Experiments demonstrate that LTGS surpasses existing baselines in reconstruction quality while enabling fast and lightweight updates.

**Strengths:**

- Proposes a new task and dataset, achieving superior performance over existing methods.

 - Employs multiple strategies to enhance temporal and multi-view consistency across different observations of the same object.

**Weaknesses:**

- Deformable objects are modeled as separate instances, leading to insufficient exploitation of temporal consistency across time.

 - Add a flowchart for Section 3.3 would help readers better understand the overall pipeline.

 - The reliance on multiple foundation models raises concerns about robustness.

**Questions:**

- How does the method perform in more complex scenes, such as when people walk in the background or curtains move due to wind? The presented examples appear static at each moment, which greatly limits the applicability of the model to dynamic real-world scenarios.

 - What level of precision is required for instance matching? How do matching errors affect reconstruction quality? Similarly, how do segmentation errors from SAM impact the final results?

 - Providing long-term reconstructed videos would help readers more effectively assess the method’s performance.

 - Will the collected real-world dataset be released publicly?

---

### Official Review · Reviewer_Q2eX · 2025-10-31

**Soundness:** 3
**Presentation:** 3
**Contribution:** 2
**Rating:** 6
**Confidence:** 5

**Summary:**

1. This paper studies long-term scene updating under sparse views. In particular, it asks how to update a 3D Gaussian scene efficiently when the scene changes, but only a few new images are available.
2. Existing methods like 4DGS mainly target continuous/deformable motions and do not handle this setting well.
3. The authors propose LTGS with three main components: (i) change detection, (ii) object template construction, and (iii) long-term Gaussian update.
4. The method is evaluated on the CL-NeRF dataset and a real dataset collected by the authors, and achieves SOTA results. Table 3 gives ablations on object tracking, pose refinement, and background initialization.

**Strengths:**

1. The problem is novel and practical: real environments change, but current methods cannot update a scene from such sparse observations.
2. The method is technically sound: it detects changes, reconstructs and stores object templates, and then updates the scene.
3. The experiments are quite thorough, especially the ablations in Table 3.

**Weaknesses:**

1. The method does not support non-rigid or articulated objects, such as people, hands, small animals, or clothes. The authors could consider integrating 4DGS-style deformation or a more general feed-forward reconstruction (e.g., VGGT-style) for local dynamic changes in the future.
2. The results seem sensitive to illumination changes, and the object templates do not appear to transfer well across scenes.
3. The motivation could be better justified: it is not fully clear what concrete real-world applications need this exact setting.

**Questions:**

Please address the issues listed in the weaknesses section.

---

### Official Review · Reviewer_HRSR · 2025-10-31

**Soundness:** 3
**Presentation:** 3
**Contribution:** 2
**Rating:** 4
**Confidence:** 4

**Summary:**

This paper tackles the problem of reconstructing scenes that undergo object-level changes over time. In particular, it presents the first approach capable of adapting to object changes and reconstructing dynamic environments even when only a sparse set of images is available at each time step. The proposed pipeline maintains previously captured Gaussian representations while detecting, tracking, and re-localizing changed objects from newly observed images. Each object is stored as an object-level Gaussian template, which is then used to refine the Gaussian parameters through optimization. By separating the dynamic objects from the static background, the framework preserves geometric consistency and integrates new object information effectively, enabling long-term scene reconstruction from sparse view updates.

**Strengths:**

1. The paper clearly defines its research problem, thoroughly analyzes the limitations of existing methods, and effectively overcomes them by combining multiple complementary techniques.
2. Unlike prior continual learning approaches based on Gaussian Splatting, the proposed framework robustly tracks object-level changes and maintains scene consistency even under sparse image updates.
3. Extensive comparisons against baselines demonstrate significant improvements in both quantitative metrics and qualitative reconstruction quality.
4. The introduction of object-level Gaussian templates as reusable structural priors enables efficient and stable long-term scene reconstruction, achieving strong generalization across multiple time steps.

**Weaknesses:**

1. The proposed method designs a pipeline that reconstructs time-varying scenes using 3D Gaussians, even with a limited number of images per time step. However, apart from the Long-Term Gaussian Splats Optimization component, most parts of the framework are built upon existing techniques, and the technical contribution is not particularly prominent. In this sense, the method appears to be a well-engineered system design that effectively integrates established modules rather than introducing fundamentally new ideas. Nonetheless, this integrative approach offers practical value and clear problem-oriented insights, though the level of research novelty may be somewhat limited.
2. The proposed method heavily relies on MASt3R for acquiring geometric information under sparse-view conditions. However, this dependence makes the overall pipeline sensitive to the performance and potential errors of MASt3R. It would strengthen the paper to include additional experiments or mechanisms demonstrating robustness to MASt3R’s estimation errors or uncertainties.

<Minor weakness>
Including a video visualization of the reconstructed 3D Gaussian templates would help clearly demonstrate the reliability of the object tracking and Gaussian update stages under dynamic scene changes.

**Questions:**

Please refer to the weaknesses mentioned above.

---

### Note · Authors · 2025-11-14

**Comment:**

We respectfully request to withdraw our submission. After further internal review, we found that our manuscript require major updates. We thank the reviewers and AC for their time and consideration.

**Withdrawal Confirmation:**

I have read and agree with the venue's withdrawal policy on behalf of myself and my co-authors.